# Strategies to Improve Adolescent Food Security from the Perspectives of Policy Advocates, Parents, and Adolescents

**DOI:** 10.3390/nu14224707

**Published:** 2022-11-08

**Authors:** Kaitlyn Harper, Rebecca Skinner, Michelle Martinez-Baack, Laura E. Caulfield, Susan M. Gross, Kristin Mmari

**Affiliations:** 1Department of Environmental Health and Engineering, Johns Hopkins Bloomberg School of Public Health, 615 N Wolfe Street, Baltimore, MD 21205, USA; 2Department of Population, Family, and Reproductive Health, Johns Hopkins Bloomberg School of Public Health, 615 N Wolfe Street, Baltimore, MD 21205, USA; 3Department of International Health, Johns Hopkins Bloomberg School of Public Health, 615 N Wolfe Street, Baltimore, MD 21205, USA

**Keywords:** food insecurity, nutritional insecurity, adolescents, federal policy

## Abstract

This study explored strategies to improve adolescent food security using semi-structured in-depth interviews with 9 policy advocates, 12 parents and 15 adolescents aged between 17 and 20 years, living in households who were eligible for the Supplemental Nutrition Assistance Program in 2020. This study was part of a larger evaluation of adolescent food insecurity conducted in Baltimore, Maryland, USA during the COVID-19 pandemic. Three key strategies arose during analysis—improving federal nutrition assistance programs for households, federal nutrition assistance programs for individual adolescents, and leveraging school programs and resources. Respondents described concordant views regarding the role of the Supplemental Nutrition Assistance Program in supporting households but held discordant views about the role of other federal programs, such as the school nutrition programs and Pandemic Electronic Benefit Transfer program. The results of this study provide important insights about policy and programmatic supports that may assist adolescents to acquire food for themselves and their families. Future research should test how federal programs and policies specifically impact food security and nutrition for adolescents.

## 1. Introduction

Food insecurity, defined as having a “lack of consistent access by all people at all times to enough food for an active, healthy life” [1], affects approximately 10% of households in the United States (US), and households with children are more likely to experience food insecurity compared to households with adults only [2]. Food insecurity is detrimental to children’s physical and psycho-social health, with potential long-term consequences [3,4]. Although parents may shield younger children from food insecurity, adolescents are typically more aware of the household’s economic environment and may take action to ensure that themselves and other members of the household have food, putting themselves at a higher risk of food insecurity [5,6]. Over 6.7 million or 16% of US adolescents live in households experiencing food insecurity [7]. In Baltimore City, the prevalence of adolescent food insecurity is higher; nearly half of adolescents report insufficient access to food [8]. Adolescents who experience food insecurity have worse physical and mental health outcomes, including higher levels of stress, depression, anxiety and panic disorders, substance use/disorders, and worse sleep outcomes [9,10,11,12,13]. Previous studies have also shown that food insecure adolescents have a higher risk of maladaptive behaviors, such as stealing, selling drugs, or selling sex [7,14].

Currently, only two previous studies have focused on understanding what strategies may improve food security specifically for adolescents [7,14]. Both qualitative studies used focus groups with adolescents to generate numerous programmatic and adolescent-targeted ideas, including combining food with other services that adolescents already attend, such as extracurricular programs and health clinics; creating more discreet ways to hand out free food, such as using different packaging or offering home delivery options; improving dissemination of information about healthy eating and assistance programs that serve adolescents; and making food more affordable to certain groups of adolescents, such as those with disabilities or those in foster care.

Adolescents also noted that improvements in the access to and quality of food in school meals programs could be a strategy for improving food security [7]. During the school year, the National School Lunch Program (NSLP) and School Breakfast Program (SBP) provide free- and reduced-price meals for children in grades K-12 who live in low-income households. During breaks, the Summer Food Service Program (SFSP) and Seamless Summer Option provide on-site congregate meals to eligible children. In some places, including Baltimore City, the Community Eligibility Provision (CEP) allows schools and/or school districts to provide free meals to all students if over 40% of the student population lives in SNAP-eligible households [15]. Although previous quantitative studies have shown that these programs are effective at reducing food insecurity in households with children [6,16,17,18], it is unclear how these programs reduce the prevalence or decrease severity of food insecurity specifically among adolescents. High school-aged students are the least likely age group in the US to utilize school meals programs (39% compared to 71% of elementary-aged students) [19]. This may be at least partially attributed to barriers such as lack of awareness, stigma, and/or negative perception of food quality [7,16].

Although most strategies described by adolescents in previous studies were individual-level strategies targeted specifically at youth, some adolescents also described a need for policy changes [7]. Adolescents described how their families did not receive enough federal nutrition assistance through programs, such as the Supplemental Nutrition Assistance Program (SNAP), to last their families the entire month. SNAP is the primary assistance program to help low-income families afford a healthy diet. Benefits are distributed to the head of household and are intended to support all members of the household; indeed, nearly half of all SNAP benefits administered are intended for children under the care of heads of households [20]. Although studies using national data have shown that receiving SNAP benefits improves household-level food security [21,22], there is mixed evidence for the effect of SNAP benefits on child food security. For example, the Summer Electronic Benefits Transfer for Children (SEBTC) pilot study found that an additional $30–$60 per child per month during the summer months was associated with lower food insecurity compared to those who did not receive the benefit [23]. Conversely, Hudak and colleagues found that an increase in the amount of SNAP benefits did not affect food security for children ages 2–18 years in low-income households [24]. Notably, neither evaluation of increased SEBTC or SNAP benefits assessed how adolescents specifically may have been impacted by these programs.

During the pandemic, Congress supported child food security and nutrition by authorizing a new federal program, Pandemic Electronic Benefits Transfer (P-EBT) [25]. This program provided households with additional benefits to purchase food for children in place of meals they may have received from school prior to the pandemic. Eligible households included those whose children received free- or reduced-price meals at school prior to the pandemic. In Baltimore, all households were eligible to receive P-EBT cards because all students are eligible to receive free school meals through CEP [15]. The program was in place nationwide for the end of the 2019–2020 school year, the entire 2020–2021 school year, and in some states, including Maryland, the summer of 2021. During this time, each child in Baltimore City automatically received a P-EBT card with their name on it through the mail or, in cases where the family already received SNAP benefits, the P-EBT amount was added to the existing card [26]. Evaluations of P-EBT found that the program significantly reduced the prevalence of households where children experienced very low food security, as reported by parents in the household [27]. However, to our knowledge, no studies have assessed the effect of P-EBT specifically on adolescent food security status.

Considering the prevalence of adolescent food insecurity and the consequences of food insecurity on current and future health, it is important to explore how strategies aimed at improving overall food security may address food insecurity among adolescents. The two previous studies conducted with adolescents provided important insights about the experiences of adolescents and their desires for change [7,14]. However, understanding the experiences, perceptions, and role of parents and policy advocates in addressing adolescent food insecurity may provide additional insights into how to better address the issue. Additionally, it is imperative to continue including adolescents in discussions about strategies that may impact them. The goal of this study was to compare proposed strategies for improving adolescent food security by three respondent groups—adolescents, parents, and policy advocates.

## 2. Materials and Methods

### 2.1. Sampling and Recruitment

This qualitative study was part of a larger evaluation of how federal nutrition assistance programs affect food security status for adolescents living in Baltimore, MD, USA during the COVID-19 pandemic. The results of this study are published elsewhere. Briefly, the study population for the larger evaluation included 289 adolescents ages 14–19 years who lived in households who were eligible to receive SNAP benefits at the time of recruitment. All adolescents in the larger study population filled out an initial survey between October 2020 and January 2021. For the study presented in this paper, recruitment was open to all respondents who filled out the initial survey. Adolescents and parents were purposively sampled by adolescent-reported food security status (i.e., food secure or food insecure), determined by their responses to the first survey.

Parents were contacted by text messages in March 2022 to ask if they would like to participate in the interviews and to ask if they would provide parental permission for their adolescents (if adolescents were under 18 years). Interviews were conducted separately, and it was not a requirement for parents and adolescents to both take part in the interviews. The sample population included both dyads, interviewed separately, as well as individual parents and adolescents. Parents and adolescents provided oral consent/assent over Zoom at the time of their respective interviews. Forty-two parents and forty adolescents were initially contacted with the goal to conduct 15 interviews and 20 interviews, respectively. Due to non-response and time constraints, 12 parents and 15 adolescents were recruited. Interviews were individually conducted over Zoom with parents and adolescents between March and May 2022.

This study utilized peer research methodology (PRM) to conduct adolescent interviews. Previous studies have shown that PRM is an important method to obtain “insider” (i.e., emic) perspective from study participants, reduce social desirability bias in participant responses, and enhance understanding of study results [28]. PRM has been used in previous research in Baltimore City to explore how adolescents experience and cope with food insecurity [14]. The study team hired two peer researchers, ages 19 and 23 years, who both grew up and lived in Baltimore their entire lives. Both peer researchers self-identified as Black/African American. Peer researchers participated in a multi-day training focused on qualitative probing techniques and Human Subjects Protections. Each received a certificate in Human Subjects Protections after completing the training. Study staff made initial connections with potential participants and received parental permission and adolescent assent, as described above. Subsequently, peer researchers reached out to potential participants via text message to schedule and conduct interviews. Peer researchers tracked their hours and received $15/hour during the training and data collection periods.

In-depth interviews were also conducted via Zoom between July and October 2021 with policy advocates, defined as individuals working at the local, state, and federal levels and who were either involved in food and nutrition policy advocacy or policymaking. Policy advocates were recruited using purposive and snowball sampling. The first two interviews were conducted with individuals identified by the study team as knowledgeable about food security policy. Potential participants were recruited via email. Each participant who finished an interview was asked if they could recommend other individuals who would be informative to talk to about the subject of food insecurity policy or adolescent food security and who potentially might also be willing participants. Snowball sampling is often used to recruit hard-to-reach populations [29]. In this study, existing participants could provide direct contact connections between researchers and potential participants. Individuals provided oral consent over Zoom at the time of the interview. A total of nine policy advocates were interviewed for this study.

### 2.2. Data Collection

#### 2.2.1. Timing of Data Collection

Data collection for this study took place within the second year of the COVID-19 pandemic. A federal status of emergency was still in place, but restrictions related to the pandemic varied by state in the US. At the time of the interviews with policy advocates, vaccines were available to residents aged 16 and over, mask mandates were still in place in Baltimore, and children and adolescents in the city had just returned to in-person school. At the time of parent and adolescent interviews, vaccinations were available for children over 6 years old and mask mandates in Baltimore City had been lifted.

#### 2.2.2. Policy Advocates

Semi-structured interviews with policy advocates were conducted before interviews with parents and adolescents. The interview guide included two main sections (Appendix A). In the first section, respondents were asked to describe causes of food insecurity generally and for adolescents. Additionally, respondents were asked to describe their opinions about how much of a problem food insecurity is for adolescents. In the second section, respondents were asked to describe strategies that would help improve food security for households in the US and specifically for adolescents. They were also asked to describe how current policies can better meet the needs of adolescents experiencing food insecurity. As is recommended in qualitative research, the interview guide was adjusted during the interview process to ensure that questions are being understood the way they were intended [30]. Additionally, themes that emerged early in the interview process were added as probes to the interview guide to provide deeper understanding of importance or unimportance. For example, one theme that emerged early on was the advantages and disadvantages of providing EBT cards to adolescents. Therefore, the idea of providing EBT cards to adolescents was added as a probe to the interview guide. Interviews lasted between 24 and 56 min, with an average length of 39 min.

#### 2.2.3. Parents

Parent interviews were conducted by three members of the study team. The interview guide was developed after the completion of policy decision maker interviews and was informed by themes that emerged during policy decision maker interviews as well as themes that emerged during previous interviews with youth as part of the parent study. There were six sections of the parent interview guide: (i) food access and food choices, (ii) parent food history, (iii) adolescent food responsibility, (iv) federal food assistance benefits during the COVID-19 pandemic; (v) food access and food security, and (vi) supporting and promoting healthy food access for adolescents (Appendix B). Parent interviews lasted between 28 and 62 min, with an average length of 42 min.

#### 2.2.4. Adolescents

All interviews with adolescents were conducted by peer researchers. Prior to data collection, two practice interviews were conducted between peer researchers and adolescents from a local nonprofit organization. After each practice interview, adult team members reviewed the recordings and transcripts and met with the peer researcher to discuss best practices for conducting interviews. After the second practice interview, the first author determined that the peer researchers were ready to conduct interviews for data collection. Each interview was conducted one-on-one with the peer researcher and the participant. Adult and peer researchers had regular check-in meetings throughout the duration of the data collection period to discuss emerging themes from the interviews, any challenges faced by the peer researchers, and adjustments needed for data collection (e.g., changes to wording of questions, addition of probes). The adolescent interview guide included five sections: (i) general questions about eating, (ii) adolescent cooking, (iii) adolescent food responsibility, (iv) federal nutrition assistance benefits, and (v) supporting and promoting healthy food access (Appendix C). Adolescent interviews lasted between 14 and 36 min, with an average length of 23 min.

### 2.3. Analysis

Each interview was transcribed verbatim using a digital transcription service (Otter.ai) and checked by simultaneously listening to the recording and reading through the transcription, correcting errors as needed. Analysis for advocates, parents, and adolescents were analyzed separately, in that order. Three members of the study team conducted inductive thematic analysis for each group of respondents, and the process was similar for all analyses. Using ATLAS.ti 9.1 (Scientific Software Development GmbH, Berlin, Germany), three members of the study team coded two transcripts independently and collaboratively to create an initial set of codes and then a set of focused codes. Coders then independently completed two additional transcripts and compared results with other coders to check for inter-rater agreement and finalize the codebook. Coders completed the remaining transcripts independently. Memo writing was used to assist the researchers with reflexivity throughout the analytic process [31].

After coding was complete, quotations from relevant codes to the research question were identified and inserted into a data matrix [32]. Matrices for policy advocates, parents, and adolescents were developed and analyzed separately. For each identified code in a matrix, the three coders read through the quotations and summarized key themes independently, then compared themes as a group. When group consensus was achieved, a final summary of each code was developed by the first author. Finally, a new matrix was created to compare summaries of all codes from all three matrices. The matrix was reviewed and discussed by the three coders and final themes were developed.

## 3. Results

### 3.1. Description of Participants

Of the nine policy advocates interviewed, five were policy advocates at the federal level; two were policy advocates at the state level; two were Maryland state senators who represent districts in Baltimore City; and one worked for the Baltimore City Health Department. All policy advocates were older than 30 years and held a bachelor’s degree or higher (i.e., Master’s Degree or Ph.D.). Of the 15 adolescents interviewed, one was 17 years, nine were 18 years old, two were 19 years and three were 20 years at the time of the interview. All five participants who were over 18 years had graduated high school and were attending college, and three of the five still lived at home. Nine of the 15 adolescents were classified as having food insecurity and four were food secure, as indicated by their scores on the second survey. Two adolescents did not fill out the second round of the survey, however, results from the first survey indicated that one had food security and the other had food insecurity. Of the 12 parents interviewed, two had children ages 17 years, seven had children ages 18 years, and three had children ages 19 years. Five parents lived in households in which the adolescent had indicated food insecurity; five parents lived in households in which the adolescents indicated food security; and two parents lived in households in which the child indicated food insecurity on the first survey but did not fill out the second survey.

The forthcoming sections outline three main themes that emerged in analysis—providing federal nutrition assistance programs to households, providing federal nutrition assistance programs directly to adolescents, and leveraging school programs and resources.

### 3.2. Providing Federal Nutrition Assistance Programs to Households

There was consensus among policy advocates, parents, and adolescents that the federal nutrition assistance programs were helpful for households and allowed them to purchase more and healthier food for adolescents. Each group also described drawbacks of programs and provided suggestions about how to improve them.

#### 3.2.1. Policy Advocates

All policy advocates were well-aware of the issue of adolescent food insecurity. They cited previous research that focused on the consequences of food insecurity, including increased risk of poor mental and physical health, as well as maladaptive coping mechanisms (e.g., stealing) used by adolescents with food insecurity to acquire food. When asked about strategies to improve food security for adolescents, policy advocates focused on federal-level policies and programs that aimed to build economic security for households with low incomes. They named federal nutrition assistance programs such as SNAP, WIC, and the Gus Schumacher Nutrition Incentive Program (GusNIP). They also named systems-focused non-nutrition policies, such as raising the minimum wage and providing a long-term child tax credit, as the most effective strategies for reducing food insecurity because they addressed the root cause of the problem—poverty. Policy advocates acknowledged that these strategies focused on households rather than specifically on adolescents and explained that addressing household-level poverty would improve the lives of all members of the household.

We know that poverty is one of the primary drivers of food insecurity. So policies that help households have more resources are all policies that we would expect would have an impact on [adolescent] food insecurity.(Federal advocate 4)

Although policy advocates were highly supportive of these programs, they also acknowledged numerous drawbacks. Almost all advocates noted that programs would be improved by increasing the amounts of benefits available to families and creating stability of programs by ensuring that families on the edge of eligibility (i.e., those who received incomes just above the eligibility requirement for the program) were able to stay on the programs despite small increases in income.

#### 3.2.2. Parents

Overwhelmingly, parents described SNAP as a helpful program that allowed them to purchase more food, and specifically more fruits and vegetables, for their children. Almost all parents brought up how the high prices of foods, especially amid rises in the previous year and throughout the COVID-19 pandemic, would have hindered their ability to purchase fruits and vegetables if they had not had SNAP. One parent explained that her adolescent daughter loved eating fruit, but she previously had to buy it in bulk at discount stores, and it did not last the entire month until the next benefits were received (as the benefits would also run out before the month’s end). Now, with the increase in SNAP benefits, she was able to purchase fresh fruit throughout the entire month.

Parents also noted that using SNAP allowed them to save money for other things they needed, such as rent or non-food items for the house. However, they also noted that as pandemic-related benefits, such as P-EBT, started to disappear, it was becoming harder to get by.

They gave us the emergency food stamps like when the kids were home. I wish they still give us that [because] it was good for the time being. […] But when they ran out, now it’s back to basics.(Parent 2)

Parents suggested that increasing the amounts of benefits long-term would help parents purchase healthier foods, which would allow adolescents and younger children in the household to have access to healthier options regularly. They also suggested changing program eligibility to allow parents who barely exceeded the income requirement to qualify. Many parents shared stories about losing their SNAP benefits when they started new or different jobs that led to increases in income, which prevented them from buying more healthy foods.

I wish they would raise [the eligibility limit] maybe… because they think that people who have made a certain amount of money, then you should be able to buy food. But when you’re buying more healthy food that takes up the cost right there.(Parent 9)

#### 3.2.3. Adolescents

Overall, adolescents were aware of the federal programs their families received and how benefits were used. Specifically, they noted that SNAP was helpful because it helped their families purchase enough food and healthier items throughout the month. Adolescents’ suggestions to improve the program mirrored advocates and parents, including improving eligibility “for families who need them” and reducing the burden of the application process. One girl with food insecurity described her experience when she helped her mom reapply for benefits.

I would make them easier to apply to and to be accepted. It’s hard to get in touch with them and hard to set up appointments. So it’s difficult to communicate with them. […] And instead of like having to constantly renew it every couple of months? [...] Like it’s just a very annoying and difficult process.(Adolescent 3)

### 3.3. Providing Federal Nutrition Assistance Programs Directly to Adolescents

Policy advocates, parents, and adolescents had widely differing views around the value of providing adolescents with their own EBT cards (as was done with P-EBT), which reflected their opinions about the role and responsibility of adolescents in the household.

#### 3.3.1. Policy Advocates

There was consensus among policy advocates that strategies targeting the household would be more effective at reducing adolescent food insecurity than strategies targeting the adolescent themselves, and strongly opposed the idea of issuing EBT cards directly to adolescents. They brought up many reasons why adolescents should not have access to their own EBT cards. For example, they described studies about adolescent food choice, noting that adolescents typically prioritize taste over healthfulness. They noted that adolescents tend to purchase unhealthy foods (i.e., those high in refined grains, sugar, and saturated fat) and that most advocates in Congress would not be supportive of providing benefits for these types of purchases. Policy advocates strongly emphasized the focus on nutrition security (i.e., consistent access to *healthy* foods) rather than simply having access to any types of foods.

So when I think of food insecurity, I’m thinking about, do residents and families have the nutritious foods and culturally relevant food that that they need in their life? It’s not about just skipping meals and getting access to anything.(City policymaker 1)

Policy advocates also mentioned that many adolescents have limited cooking abilities, and therefore may have little use for benefits, as they cannot be used to purchase prepared foods (e.g., sandwiches, fried chicken, hot pizza). They explained that there are certain groups of individuals (e.g., those experiencing homelessness, older adults) who may buy prepared foods with their benefits, but purchasing prepared foods is not as cost-effective as purchasing non-prepared foods, which makes it unfeasible at a large scale. Returning to nutrition, they also noted that there are few nutrient-dense prepared-food options available, and that adolescents would likely choose unhealthy options if they were given the choice. Additionally, policy advocates posited that adolescents should not need their own EBT card, as the amount of benefits the head of household receives on their EBT card covers the costs of food for all children in the household.

They acknowledged that there is a small portion of adolescents who may not be supported by their parents, such as situations where parents may be physically or mentally impaired. However, even in those situations, policy advocates were hesitant about giving EBT cards to adolescents because they did not want to inappropriately “adultize” adolescents by providing them the means to take on roles and responsibilities that should be reserved for parents. They reiterated that they would rather use strategies that address the entire household, because adolescent food insecurity is a symptom of household food insecurity.

I feel like that’s a step too far, too fast. I would be concerned about the unintended consequences of that. I think I’d rather be addressing the issue of like, why is that child in that situation? […] So okay, this child lives in an unstable household with a parent that’s addicted or sick or not able to provide. So all we do is give the money to the child? That doesn’t solve the problem.(Senator 2)

#### 3.3.2. Parents

Parents agreed with policy advocates that it was generally the responsibility of parents to provide for their children. Some parents noted that they would provide food for their children for as long as their children lived in the house, and others noted that they would continue to financially support their children even after they have moved out. However, respondents also emphasized the importance of adolescents having resources they could access by themselves without the help of a parent, as this takes the burden off parents, helps adolescents help themselves, and ultimately assists the family. Numerous parents brought up the advantage of providing EBT cards to adolescents. They noted that some adolescents are not fully supported by their parents and need to have access to their own card to “fend for themselves”. Unlike policy advocates, they did not describe this as a rare event. Although they noted that there were rarer circumstances in which parents were dealing with issues related to mental health or substance use, they also said that it is common for parents in low-income households to work long hours or multiple jobs to support their families. In these cases, adolescents had to take care of their own food needs.

I think [providing EBT] is more important for the young people, because we have a lot of households that are single parents and the parents are spending more time working, trying to provide a roof over their head and a warm place to stay. And a lot of times children will find themselves at home by themselves. So [children’s] ability to provide for themselves will make things a little easier for the parents, you know, who are working 12 or more hours a day, working 5 to 7 days a week trying to just trying to provide for their family, for their children.(Parent 7)

Parents employed numerous strategies to ensure their adolescents had access to resources to help them acquire food. Some parents described temporarily giving their EBT card to the adolescent so they could purchase food for themselves or the family, particularly in households with younger children. One parent mentioned giving her EBT card to her adolescent daughter in the morning before school so her daughter could purchase food on the way home. When parents received P-EBT cards during the pandemic, some managed the card similarly to their normal EBT card and others gave the card directly to the adolescent themselves, allowing them to spend it whenever and however they wanted. Parents were in consensus that P-EBT cards should be used to purchase food for their children and most noted that they allowed their children choice over what was purchased with the cards.

I would keep it, and I would only give it to them if they asked for it. Like so, can we go to the store and get some snacks? That might not be an important thing or might [not] be nourishment, but every now and again, you should treat yourself. And if they did good and they did what they were supposed to do? Absolutely. You can go get some snacks.(Parent 5)

Another strategy for improving food security that parents brought up was providing adolescents with “food vouchers”. One idea was to give adolescents vouchers for carryout places around the city where they could get a free prepared meal, with some restrictions on what they could purchase. Another idea was to provide adolescents with vouchers that would allow them to get fruits and vegetables from the supermarket or farmers market. They emphasized that it was important to give the vouchers to the adolescent themselves, so that adolescents who have responsibilities in the household would have increased resources to use without relying on their parents.

I’m thinking those programs will help teenagers because the truth of the matter is, even though [they’re] young, they’re responsible and they’re independent. You have some situations where they don’t really have a parent in the home. [...] That parent may have to work endless hours, endless hours, endless hours, and they just haven’t had a chance to go to the market.(Parent 6)

#### 3.3.3. Adolescents

Over half of adolescents in the study reported occasionally or sometimes using their parents’ EBT card to purchase food. Adolescents talked extensively about the P-EBT cards received during the pandemic. Some adolescents, particularly those experiencing food insecurity, spoke positively about P-EBT and described how helpful it was for affording healthy food during the pandemic. There was a mix of respondents who described having autonomous control over the P-EBT card and those whose parents controlled the card. Some adolescents spoke of negotiations with parents, in which the parent generally controlled the card, but the adolescent could ask for it if they wanted to buy something. Three respondents described the types of foods they purchased when using the P-EBT card on their own, and they all reported different things. One adolescent noted that they used the card to buy packaged snacks or candy for themselves, another described purchasing healthy foods for themselves, and a third described buying groceries for their family.

[I] usually go to Family Dollar, and very rarely [...] I go to places like Giant. Usually I use it for like stuff that we’re missing like, vegetables, or we’re running low on, you know, just small, like side dishes and not really main dishes.(Adolescent 3)

Adolescents provided mixed responses when asked if they would like to have their own EBT card. Some expressed wanting to have the card now, and others expressed not needing it at the time, but that they could see themselves needing it in the future, such as when they moved out of the house but were still learning how to live on their own. These responses matched their overall opinions regarding the roles and responsibilities of adolescents in the household. Most adolescents said that, ideally, it was the responsibility of the parent to provide food for their child. They noted that providing food for their children is one of the “main responsibilities as a parent”. However, they also described situations where that ideal was not possible. For example, they noted that adolescents in low-income households had more responsibility to help their families, and that adolescents with jobs should help their parents more than those without jobs.

Adolescents also spoke about their own responsibilities when there is low food availability in the house. Over half of respondents, both food-secure and food-insecure, described purchasing food for the house in times when food is low. Some respondents noted that they supported their families when they themselves had extra income to share. In these situations, they described providing money for food because they wanted to help their families. Providing food for the family was a regular occurrence in the daily lives of the adolescents; they spoke of it casually, noting multiple occasions when they cooked, provided food, or even abstained from eating as much to ensure there was enough food for the rest of the family. Respondents described taking these responsibilities on themselves, rather than having a parent or someone else in the family ask them to do it.

My parents [are] always working, so most of the time I’m home alone with my grandma and my siblings. So if there’s no food in the house and there’s nothing to eat, who’s gonna be getting all the yelling and the crying and stuff? It’s gonna be me. So I kind of have to make sure that there are foods. [...] And my grandma’s old so she can’t be doing all that stuff, so I just try to do the best I can. That’s my responsibility.(Adolescent 2)

### 3.4. Leveraging School Programs and Resources

Policy advocates, parents, and adolescents all agreed that schools were important for improving adolescent food security but had differing views on the role schools should take in the process.

#### 3.4.1. Policy Advocates

Policy advocates at the state and federal levels focused heavily on school meals as the main adolescent-targeted strategy for improving adolescent food security. They explained that implementing a policy in which all schools must serve free meals, referred to as Healthy School Meals For All (HSMA), would improve food security for children and adolescents across the US. There were a variety of reasons why they were in favor of a policy that would provide free meals for all children, but the main reason was that it would remove stigma associated with receiving free food, which is a substantial barrier to eating school meals for adolescents. Policy advocates also noted that nutrition was an important aspect of the proposed idea, and they cited studies that found that school meals were often healthier than what children ate at home. Policy advocates also described that the benefits of HSMA would extend well beyond children and adolescents to families, with both high and low incomes.

So I think if we could just normalize school meals and have it be universal school meals for all, the tide will bring up all ships, right? Like we know that that will significantly help kids in low resourced homes, and it will still benefit kids who are in higher resourced homes, because we know that school meals—when implemented correctly—are the healthiest place for kids to get at food right now.(Federal advocate 1)

However, advocates also acknowledged that numerous additional barriers prevent students from accessing consistent nutritious meals at school, such as short lunch periods and poor kitchen infrastructure. They also brought up long-standing negative perceptions of the quality of school meals as a barrier to eating the meals.

Improving the quality and the perception of school meals requires a lot of investment, but I do think it’s worthwhile. Because there are some amazing success stories. We hear a lot of the negatives […] but unfortunately, we don’t get to see the success stories a lot. And there are very progressive, well-established programs that are offering beautiful meals to kids that are healthy and colorful and culturally appropriate.(Federal advocate 3)

#### 3.4.2. Parents

Parents did not bring up the school nutrition programs as a specific solution to address food insecurity for adolescents, but around half of parents in both households with food security and food insecurity noted that their children ate school lunch on occasion. However, many parents brought up how their children avoided school meals because they did not like the taste of food, and parents had specific ideas for improving the taste, such as adding more spices to the food or buying better quality ingredients. They also suggested lengthening school mealtimes so adolescents would have enough time to finish their meals. Parents also brought up how cooking and nutrition education should be provided in schools, such as in home economics classes. They described that, although this is not a strategy for immediately accessing food, it would help adolescents choose healthier options in circumstances when they were acquiring food for themselves. This was particularly important, parents noted, for adolescents living in low-income households whose parents rely more heavily on schools to provide resources and knowledge that parents may not have the time or capacity to provide themselves. Finally, parents brought up the idea that schools could provide information about locations where adolescents would be able to get healthy food for free outside of school hours.

I think with teens, they don’t even know where to start unless somebody is providing them the information on where to get [food] from. With adults, we have more knowledge and understanding what to do, where to get it, and how to get it. But with the teens nobody’s guiding them. So how would they know where to start?(Parent 11)

#### 3.4.3. Adolescents

Adolescents had very similar ideas to parents in terms of disseminating information about free food resources. They explained that most adolescents did not know where they could access free food, and that schools would be a good place to disseminate information about food resources for adolescents, since they are accustomed to receiving information at school. They also suggested that schools would be an ideal place to hand out free food—such as fresh fruits, vegetables, canned foods, and boxed goods—for adolescents to take home at the end of the day. Adolescents distinguished these foods from the breakfasts and lunches served at school, which they often described as unappealing.

So each day a kid come to school, let them pick up a meal for dinner or something. Even if they don’t want it, you know, they can feed their family members at home who cannot afford it.(Adolescent 1)

Adolescents noted that food pantries and recreation centers might be two potential other locations that could also hand out food but affirmed that schools would be most ideal because students were already there, and it would be convenient for them to access without transportation.

## 4. Discussion

In this paper, we explored strategies to improve adolescent food security from the perspectives of federal, state, and city policy advocates, as well as parents and adolescents living in low-income urban households. Three key strategies emerged during analysis: improving federal nutrition assistance programs to households, providing federal nutrition assistance programs to individual adolescents, and leveraging school programs and resources. When discussing these strategies, respondents described concordant views regarding the role of federal programs in supporting households but held discordant views about the role of federal programs in supporting adolescents directly.

Policy advocates, parents, and adolescents shared the opinion that federal nutrition assistance programs such as SNAP positively impact food security for adolescents living in beneficiary households. Parents and adolescents noted that having SNAP benefits allowed their households to have enough food and to be able to afford healthy foods, such as more fruits and vegetables. However, all three respondent groups noted that changes, such as increasing the amounts of benefits received, expanding eligibility, and simplifying the application process, would improve households’ access to these benefits and would allow them to provide better support to all children in the household, including adolescents. These suggestions are consistent with research showing that increases in benefits and streamlining the SNAP application process during the pandemic improved food sufficiency for households with low incomes across the US [33]. Temporarily increasing benefits for households through the USDA SEBTC pilot program was also associated with lower child food insecurity, higher consumption of fruits and vegetables, whole grains, and dairy, and lower consumption of added sugar and sugar-sweetened beverages compared to households who did not receive the benefit [23]. Improving access and the amount of benefits received by families may improve food security and diet quality for adolescents and other members of the household, and future research should focus on the effects of each member of the household individually.

Results from this study and previous studies have shown that adolescents in households with low incomes often take on food-related responsibilities, such as shopping and food preparation [34,35]. However, we found that adolescents often lack support (e.g., financial resources, information on where to acquire free food, transportation to get to free food locations, or skills to prepare food once they have it) to aid them in these responsibilities. The three respondent groups in this study held discordant views about who should have direct access to federal benefits, particularly P-EBT. Typically, the amount of SNAP benefits received by a household is intended to support all members of the household, defined as “a group of people who live together and buy food and prepare meals together” [36], and respondents from all three groups acknowledged that the head of household (e.g., parents) holds the primary responsibility to provide food for children and adolescents living in the house. However, parents and adolescents in our study noted that adolescents took it upon themselves to provide food for themselves and their families, especially when parents were overburdened by working multiple jobs or long hours. Parents explained that, in those situations, it would be helpful for adolescents to have their own P-EBT cards so parents would not have to worry about their adolescent not having food. Notably, although P-EBT cards had individual children’s names on them, it was not made explicitly clear when cards were sent out if children themselves could or could not use the cards to acquire food. More research is needed to evaluate the effect of P-EBT on food security, diet quality, and food acquisition behaviors of children, adolescents, and their families.

Policy advocates were broadly against the idea of providing adolescents with their own EBT cards and described a variety of arguments against the idea, such as not wanting to adultize adolescents and the notion that adolescents would purchase unhealthy food with the card due to preferences and lack of cooking skills. The results in this study and from previous research are consistent with concerns raised by the policy advocates regarding taste preferences and cooking knowledge of adolescents [37,38,39]. However, there is currently no research on how providing direct benefits to adolescents would affect food security status or diet quality for adolescents themselves or other members of the household. Additionally, if benefits were to be distributed directly to adolescents, more research would be needed to understand implementation (i.e., would every adolescent receive a card? How would cards be distributed to reach adolescents themselves?) and potential unintended effects.

Although they viewed nutrition assistance programs as household-level solutions, policy advocates talked extensively about a different federal program that directly targets adolescents—the school meals program. They noted that providing free school meals for all children would reduce stigma for adolescents, thus improving food security. This aligns with previous research showing that adolescents in low-income households avoid eating school lunch due to stigma [7,40]. In Baltimore City, CEP was adopted starting in the 2015–2016 school year and total annual meals served increased by approximately 10% from 2015 to 2019 (subsequent years’ meal counts were affected by the COVID-19 pandemic) [41]. Previous research also found that students attending CEP schools in Baltimore City were less likely to live in food insecure households compared to students attending non-CEP schools in other districts with similar characteristics [42].

However, not everyone who has access to free school meals utilizes them. Policy advocates and parents in this study described numerous barriers related to perceptions of poor taste and quality, ability of schools to prepare meals, and short lunch periods that prevented students from eating school meals. Parent and adolescent negative perceptions of the quality of school meals has been found in previous studies [43,44]. Adolescents themselves did not mention school meals as a potential strategy to improve food access. This may be because they thought adolescents would not eat school meals due to perceptions of poor taste, as has been suggested in previous studies [45,46]. However, it is also possible that adolescents in Baltimore are accustomed to receiving free school meals and, thus, do not consider it to be a strategy for improving food access. Adolescents did note that schools would be a good place to hand out free food that students could take home after school. Currently, Feeding America—a network of 200 food banks and over 60 thousand food pantries across the US—works with the Maryland Food Bank to sponsor a School Pantry Program in 155 schools across Maryland [47]. However, it is unclear how widespread these programs are in Baltimore City high schools or the extent to which they are advertised and utilized.

Additionally, parents mentioned that having classes on cooking and nutrition in school would empower adolescents to acquire and eat healthier foods. Providing cooking classes for adolescents may address policy advocates’ concern over adolescents’ lack of cooking skills. Currently, only 7 of 155 (4.5%) schools in Baltimore City—all of which are charter or vocational—offer in-school cooking classes on a regular basis throughout the school year. Of these, five high schools offer career and technical training programs with a focus on culinary arts, and two elementary schools require cooking education classes for at least part of the year. Some schools offer after-school or sporadic in-school programming focused on cooking, but the content, extent, and reach of these programs have not been examined (Personal conversation, Baltimore City Public Schools Food and Nutrition Services, 8 December 2021). Nutrition education is taught in every grade from kindergarten through tenth grade in over 90% of US public schools, although there is no standardization across all schools with regard to the topics or comprehensiveness of curricula [48]. Only 50% of schools with grades K-8, 40% of schools with grades 9–10, and 20% of schools with grades 11–12 have district or state requirements for students to receive nutrition education [48]. Standardizing cooking and nutrition education across all grade levels in the US may support healthy eating and nutrition security for children, adolescents, and their families. Increasing cooking and nutrition knowledge offers immediate benefits and may support long-term behavior change, as behaviors and habits formed in childhood and adolescence may be carried into adulthood [49]. More research is needed on barriers and facilitators of implementing cooking and nutrition education in schools in Baltimore and across the US and its potential association with adolescent food security status.

### Limitations and Strengths

Limitations of the study included interviewing only older adolescents (ages 17–20 years). Younger adolescents may have different responsibilities related to food acquisition and may suggest other strategies for improving food security for adolescents. Additionally, due to logistical constraints, interviews with policy advocates were conducted approximately six months prior to interviews with parents and adolescents. Given the evolving federal, state, and city policies regarding the COVID-19 pandemic, respondents opinions may have changed if they were interviewed earlier (for parents and adolescents) or later (for policy advocates).

This paper also had numerous strengths. This is the first paper to bring together three groups of individuals to compare strategies to improve food security for adolescents. There is a dearth of research on the issue of adolescent food insecurity, as studies often focus on the experiences of younger children or adults. Studies that do focus on adolescents often describe the effects of food insecurity on health and behaviors, and only two studies include strategies to improve food security from the perspectives of adolescents [7,14]. This study is unique in that it includes perspectives of both parents and policy advocates, two groups who have a vested interest in improving food security for adolescents. Additionally, this study took place during the COVID-19 pandemic, which provided the opportunity to explore new federal programs, such as P-EBT, that were temporarily implemented. Although previous quantitative evaluations of P-EBT have shown that the program improves food security for households with children [27], this is the first qualitative study to explore the effect of these programs and how they may have impacted food acquisition for different members of the household. Additionally, the use of peer researchers allowed us to potentially reduce social desirability bias among adolescent respondents, enhancing the quality of our data [28].

## 5. Conclusions

This study highlighted the importance of balancing long-term and short-term solutions for improving adolescent food security and nutrition. Notably, we observed dissonance between the perceptions of adolescent food responsibilities by policy advocates and by parents and adolescents. Policy advocates focused on long-term solutions for improving food security at the household level, solutions that will address the root causes of poverty and systemic racism. Working toward these long-term solutions is crucial, and it is important to acknowledge that these solutions will require time, political will, and immense financial support. That said, there is also the need to focus on how to best support those who currently experience food insecurity and acknowledge that some members of the household—such as adolescents—might need additional support. Specific policy interventions, such as providing adolescents with their own P-EBT card, improving access to free school meals, and standardization of nutrition education, and increasing access to cooking education in grades K-12 could improve food security and nutrition. Additionally, programmatic interventions, such as providing knowledge of and access to free food resources outside of school, may assist this unique population in improving food security and nutrition for themselves and other members of their families.

This study also emphasizes the importance of accounting for lived experiences of adolescents and their families in decision making. Community ownership is important for sustainable and transformational changes to occur, and it is crucial that decision makers and others in positions of power uplift the voices of those who face historic and ongoing oppression. Adolescence is a life stage with unique challenges and opportunities, and understanding circumstances, leveraging existing assets, and addressing barriers will ultimately help improve food security and nutrition in this population.

## Data Availability

Data may be available upon request.

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
