# Peer review of "Strategies to Improve Adolescent Food Security from the Perspectives of Policy Advocates, Parents, and Adolescents"

_nutrients, 2022, doi:10.3390/nu14224707_

Round 1
Reviewer 1 Report
The article is interesting but requires:
-Include summarized and key aspects of the design of the surveys in a table.
- Include global results in numerical form, ideally in the form of graphs.
- Is it possible to obtain statistically valid results from the study population with such a low number of responses? Indicate and comment in the text on the statistical representativeness aspects included in the study design.
Author Response
We appreciate the reviewer taking the time to read our article and provide feedback. Please see the responses to each comment below.
1. Include summarized and key aspects of the design of the surveys in a table.
Thank you for your comment. The survey briefly described in the methods was simply used for recruitment of adolescent and parent participants. In other words, the survey sample was used only as the sampling frame for the purposive sampling of participants. The quantitative results of this study, including results from the survey, are currently in preparation. For this qualitative study, which does not include any quantitative results, we believe it would be beyond the scope of the paper to provide further information than that presented. We have reviewed the manuscript to make sure we are clear on this point.
2. Include global results in numerical form, ideally in the form of graphs.
We appreciate the reviewer’s commitment to quantitative rigor; however, we believe that numerical representation of our results is not necessary to our key results or overall message. This paper presents the results from a qualitative study, defined by Fossey and colleagues as a set of “research methodologies that describe and explain persons’ experiences, behaviors, interactions and social contexts without the use of statistical procedures or quantification” (2002). The theoretical basis, methods, presentation of results, and interpretation of results differ between qualitative and quantitative research. Although qualitative results are often presented in a similar manner to quantitative (i.e., in a results section, followed by a discussion section), they do not require tables or figures in their presentation. Conversely, themes are often described and accompanied by appropriate verbatim quotes to illustrate findings (Burnard et al., 2008). For these reasons, we do not believe the inclusion of numerical results is appropriate in our research and have not made any changes in response to this comment.
Fossey E, Harvey C, McDermott F, Davidson L. Understanding and evaluating qualitative research. Australian & New Zealand journal of psychiatry. 2002 Dec;36(6):717-32. https://doi.org/10.1046/j.1440-1614.2002.01100.x
Burnard P, Gill P, Stewart K, Treasure E, Chadwick B. Analysing and presenting qualitative data. British dental journal. 2008 Apr;204(8):429-32. https://doi.org/10.1038/sj.bdj.2008.292
3. Is it possible to obtain statistically valid results from the study population with such a low number of responses? Indicate and comment in the text on the statistical representativeness aspects included in the study design.
Were this a quantitative study, we would agree that sample size should be accounted for in terms of validity. As this is a qualitative study, our sampling was not designed to draw population-based inferences. Since we did not ask the same questions in the same way of every respondent, there is no true denominator. Thus, reporting statistical results would have no real meaning. For similar reasons, most qualitative studies avoid presenting percentages or proportions. We believe that doing so here would be misleading to the reader, creating the false impression that we are making population-based inferences when the study was not intended for that purpose. We did not make changes in response to this comment.
Reviewer 2 Report
The research question is not new, but the authors approach it from an interesting perspective, adding novelty to the manuscript. Here are my comments:
- I would see a table which describes the respondents' socio-economic characteristics (e.g., age, gender, education, BMI, etc,)
- A large part of the conclusions is not appropriate, as it is only a concise repetition of the comments. In the conclusions, you should simultaneously consider all you have discovered, and exploit it to add something new (or new interpretations), and policy indications.
- The authors do not discuss possible limitations of their study or the insights for future directions of research. Maybe they could discuss external validity of the results in terms of possible insights in other countries and/or additional variables that they would have liked to have to better answer to their research question. I suggest expanding the conclusions section.
Author Response
A big thank you to the reviewer for reading and providing comments on our paper. Please see our responses to each comment below.
1. I would [like to] see a table which describes the respondents' socio-economic characteristics (e.g., age, gender, education, BMI, etc,)
Thank you for your comment. We did not include a table of socio-economic or demographic characteristics in this paper because—other than age, which is reported in Section 3.1—they are not central to our research question. Additionally, we collected socio-economic and demographic characteristics only for adolescents in the study, and it may be confusing to the reader to see characteristics for one of the three sample groups. Notably, the quantitative results of the parent study, including survey results, are currently in preparation.
2. A large part of the conclusions is not appropriate, as it is only a concise repetition of the comments. In the conclusions, you should simultaneously consider all you have discovered, and exploit it to add something new (or new interpretations), and policy indications.
Thank you for your comment. We have added an additional paragraph in the discussion that adds new ideas regarding the importance of including community voice (lines 689-695). Additionally, we have expanded on our ideas by providing clear examples for solutions that are both policy and programmatic in nature (683-686).
3. The authors do not discuss possible limitations of their study or the insights for future directions of research. Maybe they could discuss external validity of the results in terms of possible insights in other countries and/or additional variables that they would have liked to have to better answer to their research question. I suggest expanding the conclusions section.
Thank you for your comment. In Section 4.1 (Strengths and Limitations), we highlight the following potential limitations of our study:
Limitations of the study included interviewing only older adolescents (ages 17-20 years). Younger adolescents may have different responsibilities related to food acquisition and may suggest other strategies for improving food security for adolescents. Additionally, due to logistical constraints, interviews with policy advocates were conducted approximately six months prior to interviews with parents and adolescents. Given the evolving federal, state, and city policies regarding the COVID-19 pandemic, respondents opinions may have changed if they were interviewed earlier (for parents and adolescents) or later (for policy advocates).
Additionally, in Section 4 (Discussion), we describe numerous areas in which future research is necessary, including evaluation of the effect of P-EBT on food security, diet quality, and food acquisition behaviors of children, adolescents, and their families and the effect of providing direct benefits to adolescents on food security status or diet quality for adolescents themselves or other members of the household, as well as implementation research and unintended consequences. We have also provided some examples of gaps in research, including how widespread cooking/nutrition programs are in Baltimore City high schools or the extent to which they are advertised and utilized. We have clarified for the reader some of these gaps with specific language around where future studies may be necessary (lines 563-564, 644-646).
Round 2
Reviewer 2 Report
I am currently fine with the revised manuscript version. Authors properly addressed referee's comments.